# Neurosymbolic Grounding for Compositional World Models

**Atharva Sehgal**
UT Austin

**Arya Grayeli**
UT Austin

**Jennifer J. Sun**
Caltech

**Swarat Chaudhuri**
UT Austin

## Abstract

We introduce COSMOS, a framework for object-centric world modeling that is designed for compositional generalization (CompGen), i.e., high performance on unseen input scenes obtained through the composition of known visual "atoms." The central insight behind COSMOS is the use of a novel form of neurosymbolic grounding. Specifically, the framework introduces two new tools: (i) neurosymbolic scene encodings, which represent each entity in a scene using a real vector computed using a neural encoder, as well as a vector of composable symbols describing attributes of the entity, and (ii) a neurosymbolic attention mechanism that binds these entities to learned rules of interaction. COSMOS is end-to-end differentiable; also, unlike traditional neurosymbolic methods that require representations to be manually mapped to symbols, it computes an entity's symbolic attributes using vision-language foundation models. Through an evaluation that considers two different forms of CompGen on an established blocks-pushing domain, we show that the framework establishes a new state-of-the-art for CompGen in world modeling. Artifacts are available at `https://trishullab.github.io/cosmos-web/`.

## 1 Introduction

The discovery of world models — deep generative models that predict the outcome of an action in a scene made of interacting entities — is a central challenge in contemporary machine learning (Ha & Schmidhuber, 2018; Hafner et al., 2023). As such models are naturally factorized by objects, methods for learning them (Zhao et al., 2022; Goyal et al., 2021; Watters et al., 2019a; Kipf et al., 2019) commonly follow a modular, object-centric perspective. Given a scene represented as pixels, these methods first extract representations of the entities in the scene, then apply a transition network to model interactions between the entities. The entity extractor and the transition model form an end-to-end differentiable pipeline.

Of particular interest in world modeling is the property of *compositional generalization* (CompGen), i.e., test-time generalization to scenes that are novel compositions of known visual "atoms". Recently, Zhao et al. (2022) gave a first approach to learning world models that compositionally generalize. Their method uses an *action attention* mechanism to bind actions to entities. The mechanism is equivariant to the replacement of objects in a scene by other objects, enabling CompGen.

This paper continues the study of world models and compositional generalization. We note that such generalization is hard for purely neural methods, as they cannot easily learn encodings that can be decomposed into well-defined parts. Our approach, COSMOS, uses a novel form of *neurosymbolic grounding* to address this issue.

The centerpiece idea in COSMOS is the notion of *object-centric, neurosymbolic scene encodings*. Like in prior modular approaches to world modeling, we extract a discrete set of entity representations from an input scene. However, each of these representations consists of: (i) a standard real vector representation constructed using a neural encoder, and (ii) a vector of *symbolic attributes*, capturing important properties — for example, shape, color, and orientation — of the entity.

Like Goyal et al. (2021), we model transitions in the world as a collection of neural modules, each capturing a "rule" for pairwise interaction between entities. However, in contrast to prior work, we bind these rules to the entities using a novel *neurosymbolic attention* mechanism. In our version of

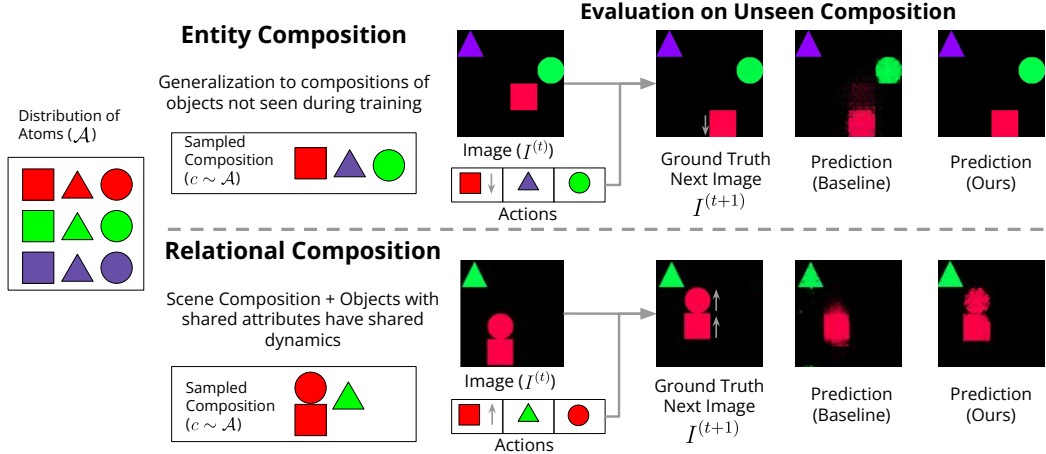

Figure 1: Overview of compositional world modeling. We depict examples from a 2D block pushing domain consisting of shapes that interact, where we can generate samples of different shapes and interactions. We aim to learn a model that generalizes to compositions not seen during training, such as entity composition (*top*) and relational composition (*bottom*). Previous works (Goyal et al., 2021) focus on entity composition, and struggle to generalize to harder compositional environments. Our approach COSMOS leverages object-centric, neurosymbolic scene encodings to compositionally generalize across settings containing different types of compositions.

such attention, the keys are symbolic and the queries are neural. The symbolic keys are matched with the symbolic components of the entity encodings, enabling decisions such as "the $i$-th rule represents interactions between a black object and a circular object" (here, "black" and "circular" are symbolic attributes). The rules, the attention mechanism, and the entity extractor constitute an end-to-end differentiable pipeline.

The CompGen abilities of this model stems from the compositional nature of the symbolic attributes. The symbols naturally capture "parts" of scenes. The neurosymbolic attention mechanism fires rules based on (soft, neurally represented) logical conditions over the symbols and can cover new compositions of the parts.

A traditional issue with neurosymbolic methods for perception is that they need a human-provided mapping between perceptual inputs and their symbolic attributes (Harnad, 1990; Tang & Ellis, 2023). However, COSMOS automatically obtains this mapping from vision-language foundation models. Concretely, to compute the symbolic attributes of an object, we utilize CLIP (Radford et al., 2021) to assign each object values from a set of known attributes.

We compare COSMOS against a state-of-the-art baseline and an ablation on an established domain of moving 2D shapes (Kipf et al., 2019). Our evaluation considers two definitions of CompGen, illustrated in Figure 1. In one of these, we want the model to generalize to new *entity compositions*, i.e., input scenes comprising sets of entities that have been seen during training, but never together. The other definition, *relational composition*, is new to this paper: here, we additionally need to accommodate shared dynamics between objects with shared attributes (e.g., the color red). Our results show that COSMOS outperforms the competing approaches at next-state prediction (Figure 1 visualizes a representative sample) and separability of the computed latent representations, as well as accuracy in a downstream planning task. Collectively, the results establish COSMOS as a state-of-the-art for CompGen world modeling.

To summarize our contributions, we offer:

- COSMOS, the first differentiable neurosymbolic approach — based on a combination of neurosymbolic scene encodings and neurosymbolic attention — to object-centric world modeling.

- a new way to use foundation models to automate symbol construction in neurosymbolic learning;

- an evaluation that shows COSMOS to produce significant empirical gains over the state-of-the-art for compositionally generalizable world modeling.

## 2 PROBLEM STATEMENT

We are interested in learning world models that compositionally generalize. World models arise naturally out of the formalism for Markov decision processes (MDPs). An MDP $\mathcal{M}$ is a tuple $(\mathcal{S}, \mathcal{A}, T, R, \gamma)$ with states $\mathcal{S}$, actions $\mathcal{A}$, transition function $T : \mathcal{S} \times \mathcal{A} \to \mathcal{S} \times \mathbb{R}_{\geq 0}$, reward function $R : \mathcal{S} \times \mathcal{A} \to \mathbb{R}$, and discount factor $\gamma \in [0, 1]$. We make three additional constraints:

**Object-Oriented state and action space**: In our environments, the state space is realized as images. At each time step $t$, an agent observes an image $I \in \mathcal{S}_{pixel}$ and takes an action $A \in \mathcal{A}$. However, learning $\mathcal{S}_{pixel}$ directly is an intractable problem. Instead, we assume that the high dimensional state space can be decomposed into a lower dimensional *object-oriented* state space $\mathcal{S} = \mathcal{S}_1 \times \cdots \times \mathcal{S}_k$ where $k$ is the number of objects in the image. Now, each *factor* $S_i \in \mathcal{S}_i$ describes a single object in the image. Hence, the transition function has signature $T : (S_1; A_1 \times \cdots \times S_k; A_k) \to (S_1 \times \cdots \times S_k)$ where each factor $A_i$ is a factorized set of actions associated with object representation $S_i$ and $(\circ; \circ)$ is the concatenation operator. A pixel grounding function $P_{\downarrow} : \mathcal{S} \to \mathcal{S}_{pixel}$ enables grounding a factored state into an image.

**Symbolic object relations**: We assume that objects in the state space share attributes. An attribute is a set of unique symbols $C_p = \{C_p^1, \ldots C_p^q\}$. For instance, in the 2D block pushing domain (illustrated in Figure 1), each object has a "color" attribute that can take on values $C_{\texttt{color}} := \{\texttt{red}, \texttt{green}, \ldots\}$. An object can be composed of many such attributes that can be retrieved using an attribute projection function $\alpha_{\uparrow} : S_i \to \Lambda_i := (\Lambda_i^{C_1}; \ldots; \Lambda_i^{C_p})$ where $(C_1, \ldots, C_p)$ is a predefined, ordered list of attributes and $\Lambda_i^{C_p}$ selects the value in the $C_p$-th attribute that is most relevant to $S_i$. Note that $\alpha_{\uparrow}$ only depends on a single object and trivially generalizes to different compositions of objects (Keysers et al., 2020).

**Compositional Generalization**: We assume that each state in our MDP can be decomposed using two sets of elements: compounds and atoms. *Compounds* are sets of elements that can be decomposed into smaller sets of elements, and *atoms* are sets of elements that cannot be decomposed further. For instance, in the block pushing domain (Figure 1), we can designate each unique object as an atom and designate the co-occurrence of a set of atoms in a scene as a compound. We use $\texttt{A}$ to denote the atoms and $\texttt{C}$ to denote the compounds. The frequency distribution of a distribution $\circ$ is denoted $\mathcal{F}_{\circ}(\mathcal{D})$. Given this, CompGen is expressed as a property of the train distribution $\mathcal{D}_{train}$ and of the test distribution $\mathcal{D}_{test}$ undergoing a distributional shift of the compounds, while the distribution of atoms remains the same.

Given the following assumptions, for any experience buffer $\mathcal{D} := \{\{\langle I^{(t)}, A^{(t)} \rangle\}_{t=1}^T \subseteq \mathcal{S}_{pixel} \times \mathcal{A}\}_{i=1}^N$, learning a world model boils down to learning the transition function $T$ for the MDP using the sequences of observations collected by $\mathcal{D}$. Specifically, for $S_{1 \ldots k}^{(i,t)} \in I^{(i,t)}, A^{(i,t)}, I^{(i,t+1)} \in \mathcal{D}$, our objective function is

$$\mathcal{L} = \frac{1}{N(T-1)} \sum_{i=1}^{N} \sum_{t=1}^{T-1} \|P_{\downarrow}\left(T\left(S_1^{(i,t)}; A_1^{(i,t)} \ldots, S_k^{(i,t)}; A_k^{(i,t)}\right)\right) - I^{(i,t+1)}\|_2^2$$

We study two kinds of compositions:

**Entity Composition**: Figure 1 shows an instance of entity centric CompGen in the block pushing domain. Here, there exist $n = 9$ unique objects, but only $k = 3$ are allowed to co-occur in any given realized scene. Each unique object represents an atom, and the co-occurrence of a set of $k$ atoms in a scene represents the compound. Hence, there are a total $\binom{n}{k}$ possible compositions. The distribution of atoms in the train distribution and the test distribution does not change, while the distribution of compounds at train time and at test time are disjoint. So, $\mathcal{F}_{\texttt{A}}(\mathcal{D}_{eval})$ is equivalent to $\mathcal{F}_{\texttt{A}}(\mathcal{D}_{train})$ while $\mathcal{F}_{\texttt{C}}(\mathcal{D}_{eval}) \cap \mathcal{F}_{\texttt{C}}(\mathcal{D}_{train}) = \varnothing$.

**Relational Composition**: Figure 1 shows an instance of relational composition in the block pushing domain. Here, there are $n = 9, k = 3$ unique objects. A composition of objects occurs when two objects share attributes (for instance, here, the shared attributes are color and adjacency). Objects with shared attributes share dynamics. As a result, if one object experiences a force, others with the same attributes also undergo that force. Hence, assuming a single composition per scene, there are a total $\binom{n}{k}\binom{k}{2}$ possible compositions. As in the previous case, $\mathcal{F}_{\texttt{A}}(\mathcal{D}_{eval})$ is equivalent to $\mathcal{F}_{\texttt{A}}(\mathcal{D}_{train})$ while $\mathcal{F}_{\texttt{C}}(\mathcal{D}_{eval}) \cap \mathcal{F}_{\texttt{C}}(\mathcal{D}_{train}) = \varnothing$.

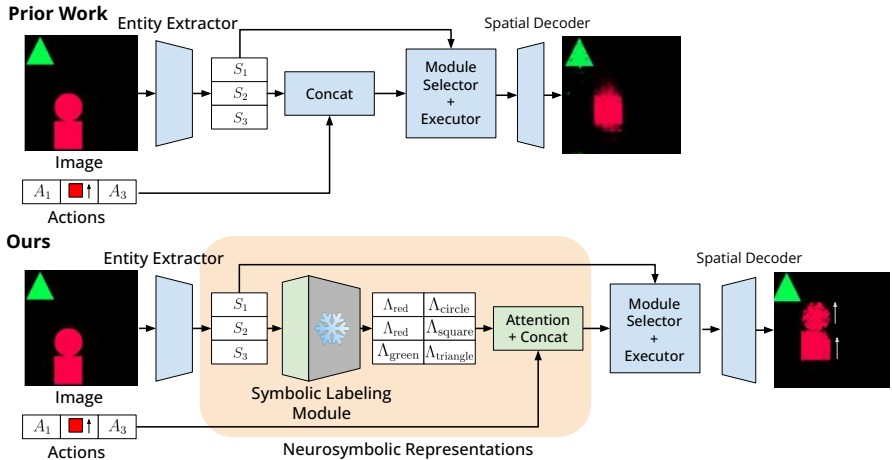

Figure 2: Comparing world modeling frameworks between prior work (Goyal et al., 2021) and Cos-MOS. Both modules start with entity extraction, to obtain neural object representations $\{S_1, \ldots S_k\}$ from the image (Section 3.1). While prior work uses this representation directly for the module selector, our work leverages a symbolic labeling module, which outputs a set of attributes $\Lambda$, to learn neurosymbolic representations (Section 3.2). We then perform action conditioning (Section 3.2) to keep track of corresponding actions, and update through a transition model (Section 3.3).

# 3 METHOD

We approach neurosymbolic world modeling through object-centric neurosymbolic representations (Figure 2). Our method consists of: extracting object-centric encodings (Section 3.1), representing each object using neural and symbolic attributes (Section 3.2), and learning a transition model to update to next state (Section 3.3). Similar to previous works, we use slot based autoencoders to extract objects, but use symbolic grounding from foundation models alongside neural representations, which enables our model to achieve both entity and relational compositionality. The full algorithm is presented in Algorithm 1, and the model is visualized in Figure 3. We will make use of the example in Figure 3 to motivate our algorithm.

---

**Algorithm 1** Neurosymbolic world model for $k$-factorized state space. The input is an image $I$ of dimensions (C, H, W) = $(3 \times 224 \times 224)$ and a set of $k$-factorized actions of dimension $(k \times d_{\texttt{action}})$. The model is trained end-to-end except for the SLM modules, whose weights are held constant. There are three global variables: $T$, $\{C_1, \ldots C_p\}$ and $\{\vec{R}_1, \ldots \vec{R}_l\}$. $T$ is the threshold of the number of repeated slot update steps, $\{C_1, \ldots C_p\}$ denotes text encodings for the closed vocabulary and $\{\vec{R}_1, \ldots \vec{R}_l\}$ are learnable rule encodings.

---

1: **function** TRANSITIONIMAGE($I, [A_1, \ldots \ldots A_k]$)
2:     $\{S_1, \ldots S_k\} \leftarrow$ ENTITYEXTRACTOR($I$)         $\triangleright S_i$ dim: $(k, d_{\texttt{slot}})$
3:     **for** _ **in** range($T$) **do**
4:        $\{I'_1, \ldots I'_k\}, \{M'_1, \ldots M'_k\} \leftarrow$ SPATIALDECODER($\{S_1, \ldots S_k\}$)      $\triangleright I'_i$ dim: (C, H, W)
5:        $\{I_1, \ldots I_k\} \leftarrow \{M'_i \cdot I'_i, \forall i \in [1, k]\}$
6:        $\{\Lambda_1, \ldots \Lambda_k\} \leftarrow \{$SLM($I_i, \{C_1, \ldots C_p\}$)$|\forall i \in [1, k]\}$         $\triangleright$ dim: $(k, p)$
7:        $\{\overline{\Lambda}_1, \ldots \overline{\Lambda}_k\} \leftarrow$ ACTIONATTN($\{\Lambda_1, \ldots \Lambda_k\}, [A_1, \ldots A_k]$)
8:        $p, c, r \leftarrow$ MODULARRULENET(K=$\{\overline{\Lambda}_1, \ldots \overline{\Lambda}_k\}$, Q=$\{\vec{R}_1, \ldots \vec{R}_l\}$)
9:        $S_p \leftarrow S_p +$ MLPBANK[$r$](**concat**($S_p, S_c, \vec{R}_r$))
10:     $\{I'_1, \ldots I'_k\}, \{M'_1, \ldots M'_k\} \leftarrow$ SPATIALDECODER($\{S_1, \ldots S_k\}$)      $\triangleright I'_i$ dim: (C, H, W)
11:     **return** $\sum_{i=1}^k M'_i \cdot I'_i$

---

## 3.1 SLOT-BASED AUTOENCODER

A slot-based autoencoder transforms an image into a factorized hidden representation, with each factor representing a single entity, and then reconstructs the image from this representation. Such

autoencoder make two assumptions: (1) each factor captures a specific property of the image (2) collectively, all factors describe the entire input image. This set-structured representation enables unsupervised disentanglement of objects into individual entities. Our slot based autoencoder has two components:

ENTITYEXTRACTOR : $I \to \{S_1, \ldots, S_k\}$: The entity extractor takes an image and produces a set-structured hidden representation describing each object. In our domains, slot attention and derivate works (Chang et al., 2022; Jia et al., 2022) struggle to disentangle images into separate slots. To avoid the perception model becoming a bottleneck for studying dynamics learning, we propose a new entity extractor that uses Segment Anything Kirillov et al. (2023) with pretrained weights to produce set-structured segmentation masks to decompose the image into objects and a Resnet-18 image encoder He et al. (2016) to produce a set-structured hidden representation for each object. This model allows us to perfectly match ground truth segmentations in our domains while preserving the assumptions of set structured hidden representations. More details are presented in Section A.1.

SPATIALDECODER : $\{S_1, \ldots, S_k\} \to \{I'_1, \ldots I'_k\}, \{M'_1, \ldots M'_k\}$: The slot decoder is a permutation equivariant network that decodes the set of slots back to the input image. We follow previous works (Locatello et al., 2020; Zhao et al., 2022; Goyal et al., 2021) in using a spatial decoder (Watters et al., 2019b) that decodes a given set of vectors $\{S_1, \ldots, S_k\}$ individually into a set of image reconstructions $\{I'_1, \ldots I'_k\}$ and a set of mask reconstructions $\{M'_1, \ldots, M'_k\}$. The final image $I$ is produced by taking the Hadamard product of each reconstruction and its corresponding mask and adding all the resulting images together. That is, $I = \sum_{i=1}^{k} M'_i \cdot I'_i$.

## 3.2 NEUROSYMBOLIC ENCODING

To achieve robust CompGen in our setting, our representation must be resilient to both object and attribute compositions. The $k$ slot-based encoding, by construction, generalizes to object replacement. For attribute compositions, however, it is essential to know the exact attributes to be targeted for CompGen. Given these attributes, we propose describing each entity with a composition of *symbol vectors*. Each symbol vector is associated with a single entity and attribute, allowing it to trivially generalize to different attributes compositions. Moreover, we can ensure a canonical ordering of the symbols, making downstream attention-based computations invariant to permutations of attributes. We'll next detail our method for generating these symbol vectors.

SLM : $I_i \times \{C_1, \ldots C_p\} \to \Lambda_i := (\Lambda_i^{C_1}; \ldots; \Lambda_i^{C_p})$: The symbolic labelling module (SLM) processes an image and a predefined list of attributes. Assuming this list is comprehensive (though not exhaustive), the module employs a pretrained CLIP model to compute attention scores between the image features and each entity encoding. The resulting logits indicate the alignment between the image and each attribute. The attribute most aligned with the image is then identified using a straight-through Gumbel softmax (Jang et al., 2016). The gumbel-softmax yields the index of the most likely value for each attribute as one-hot vectors, which are concatenated together to form a bit-vector. However, the discrete representation of such bit-vectors do not align well with downstream attention based modules so, instead of directly using one-hot vectors, the gumbel-softmax selects a value-specific encoding for each attribute. Thus, the resultant symbol vector is a composition of learnable latent encodings distinct to each attribute value. In implementation, SLM utilizes another zero-shot symbolic module (a spatial-softmax operation (Watters et al., 2019b)) to extract positional attributes, such as the $x$ and $y$ position of the object, from the disentangled input vector.

For instance, in Figure 3, SLM takes in the slot corresponding to the circle and a list of attributes (shape, color, etc.) and selects the most relevant attribute value ('circle', 'red', etc.). We subsequently select trainable embeddings corresponding to each attribute value and concatenate them to construct the symbolic embedding.

ACTIONATTN : $\{\Lambda_1, \ldots \Lambda_k\} \times \{A_1, \ldots A_k\} \to \{\overline{\Lambda}_1, \ldots, \overline{\Lambda}_k\}$: For accurate next state prediction, world models must condition the state on the corresponding action. Typically, this is done by concatenating each action to its associated encoding if the slots have a canonical ordering. However, the entities in a set structured representation do not have a fixed order. To find such a canonical ordering, we follow Zhao et al. (2022) in learning a permutation matrix between the actions and the slots and using the permutation matrix to reorder the slots and concatenate them with their respective actions. Attention, by construction, is equivariant with respect to slot order, which avoids the need

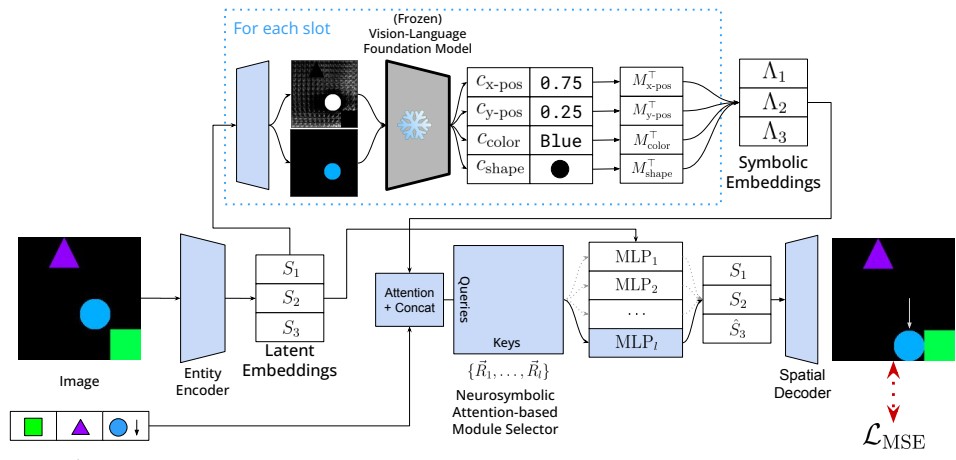

Figure 3: A single update step of COSMOS. The image $I$ is fed through a slot-based autoencoder and a CLIP model to generate the slot encodings $\{S_1, \ldots S_k\}$ and the symbol vectors $\{\Lambda_1, \ldots \Lambda_k\}$. The actions and the symbolic encoding are aligned and concatenated using a permutation equivariant action attention module, which are used to select the update rule to be applied to the slots. This figure depicts a single update step; in implementation, the update-select-transform step is repeated multiple times to model multi-object interactions.

to enforce a canonical ordering. For instance, in Figure 3, if $\Lambda_1$ corresponds to the circle, a trained ACTIONATTN module reorders the actions so that $A_1$ corresponds to going downwards.

### 3.3 TRANSITION MODEL

Monolithic transition models like GNNs and MLPs model every pairwise object interaction, leading to spurious correlations in domains with sparse interactions. Our modular transition model addresses this problem by selecting a relevant pairwise interaction and updating the encodings for those objects locally (following Goyal et al. (2021)). We model this selection process by computing key-query *neurosymbolic attention* between ordered symbolic and neural rule encodings to determine the most applicable rule-slot pair. As each entity is composed of shared symbol vectors, the dot-product activations between the symbolic encodings and the rule encodings remain consistent for objects with identical attributes. Next, we will discuss the mechanisms of module selection and transitions.

MODULESELECTOR : $\{\overline{\Lambda}_1, \ldots \overline{\Lambda}_k\} \times \{\vec{R}_1, \ldots \vec{R}_l\} \rightarrow (p, c, r)$: The goal of the selection process is to select the primary slot, the contextual slot (which the primary slot will be conditioned on), and the update function to model the interaction. Query-key attention (Bahdanau et al., 2015) serves as the base mechanism we will be using to perform selections. We compute query-key attention between the rule encoding and the action-conditioned symbolic encoding. The naive algorithm to compute this selection will take $O(k^2 l)$ time, where $k$ is the number of slots and $l$ is the number of rules. In implementation, the selection of $(p, c, r)$ can be reduced to a runtime of $O(kl + k)$ by *partial application* of the query-key attention. This algorithm is presented in Section 2 of the appendix.

MLPBANK : $i \rightarrow \text{MLP}_i$: Our modular transition function comprises a set of rules $\mathbf{R}_1, \ldots, \mathbf{R}_n$, with each rule defined as $\mathbf{R}_i = (\vec{R}_i, \text{MLP}_i)$. Here, $\vec{R}_i$ is a learnable encoding, while $\text{MLP}_i : S_p \times S_c \rightarrow S_p'$ represents a submodule that facilitates pairwise interactions between objects. Intuitively, each submodule is learning how a primary state changes conditioned on a secondary state. In theory, each update function can be customized for different problems. However, in this study, we employ multi-layered perceptrons for all rules.

## 4 EXPERIMENTS

We demonstrate the effectiveness of COSMOS on the 2D Block pushing domain (Kipf et al., 2019; Zhao et al., 2022; Goyal et al., 2021; Ke et al., 2021) with entity composition, and two instances of relational composition. We selected the 2D block pushing domain as it is a widely-adopted and

| Dataset | Model | 3 objects | | | 5 objects | | |
|---------|-------|-----------|---|---|-----------|---|---|
| | | MSE ↓ | AE-MSE ↓ | Eq.MRR ↑ | MSE ↓ | AE-MSE ↓ | Eq.MRR ↑ |
| RC (Sticky) | COSMOS | **4.23E-03** | **4.90E-04** | **1.20E-01** | **4.15E-03** | **1.68E-03** | **3.67E-01** |
| | ALIGNEDNPS | 1.14E-02 | 7.72E-03 | 8.01E-02 | 6.07E-03 | 2.47E-03 | 3.62E-01 |
| | GNN | 7.94E-03 | 5.11E-03 | 6.03E-04 | 6.21E-03 | 2.73E-03 | 5.30E-04 |
| RC (Team) | COSMOS | **4.60E-03** | **4.33E-04** | 1.04E-01 | **5.53E-03** | 1.86E-03 | 2.86E-01 |
| | ALIGNEDNPS | 1.24E-02 | 8.36E-03 | **1.75E-01** | 9.64E-03 | 3.12E-03 | **2.93E-01** |
| | GNN | 8.92E-03 | 3.82E-03 | 7.16E-04 | 7.01E-03 | **1.62E-03** | 5.46E-04 |
| EC | COSMOS | **7.66E-04** | **6.34E-05** | 2.99E-01 | **4.08E-04** | **2.92E-06** | 3.03E-01 |
| | ALIGNEDNPS | 3.51E-03 | 2.69E-03 | 2.97E-01 | 2.45E-03 | 1.22E-03 | 3.19E-01 |
| | GNN | 9.89E-03 | 1.03E-02 | **5.50E-01** | 1.20E-02 | 1.28E-02 | **5.25E-01** |

Table 1: Evaluation results on the 2D block pushing domain for entity composition (EC) and relational composition (RC) averaged across three seeds. We report next-state reconstruction error (MSE), autoencoder reconstruction error (AE-MSE), and the equivariant mean reciprocal rank (Eq.MRR) for three transition models: our model (COSMOS), an improved version of Goyal et al. (2021) (ALIGNEDNPS), and a reimplementation of Zhao et al. (2022) (GNN). Our model (COSMOS) achieves best next-state reconstructions for all datasets.

well-studied benchmark in the community. Notably, even in this synthetic domain, our instances of compositional generalization proved challenging to surpass for baseline models, underscoring the difficulty of the problem.

**2D Block Pushing Domain**: The 2D block pushing domain (Kipf et al., 2019; Ke et al., 2021) is a two-dimensional environment that necessitates dynamic and perceptual reasoning. Figure 5 presents an overview of this domain. All objects have four attributes: color ($\Lambda^{C_{\text{color}}}$), shape ($\Lambda^{C_{\text{shape}}}$), $x$ position ($\Lambda^{C_{\text{x-pos}}}$), and $y$ position ($\Lambda^{C_{\text{y-pos}}}$). Objects can be pushed in one of the four cardinal directions (North-East-South-West). Heavier objects can push lighter objects, but not the other way around. The weight of the object depends on the shape attribute. At each step, the agent observes an image of size $3 \times 224 \times 224$ with $k$ objects and an action pushing one of the objects. This action is chosen from a uniform random distribution. The goal is for the agent to capture the dynamics of object movement. Furthermore, while there are $n = |A_{\text{color}}| \times |A_{\text{shape}}|$ unique objects in total, only $k < n$ objects are allowed to co-occur in a realized scene.

**Dataset Setup**: We adapt the methodology from (Kipf et al., 2019; Zhao et al., 2022) with minor changes. For entity compositions (EC), we construct training and testing datasets to have unique object combinations between them. In relational composition (RC), objects with matching attributes exhibit identical dynamics. Two specific cases are explored: Team Composition (RC-Team) where dynamics are shared based on color, and Sticky Composition (RC-Sticky) where dynamics are shared based on color and adjacency. Further details are in the appendix (Section A.2). Our data generation methodology ensures that the compound distribution is disjoint, while the atom distribution remains consistent across datasets, i.e. $\mathcal{F}_{\text{C}}(\mathcal{D}_{train}) \cap \mathcal{F}_{\text{C}}(\mathcal{D}_{eval}) = \emptyset$ and $\mathcal{F}_{\text{A}}(\mathcal{D}_{train}) = \mathcal{F}_{\text{A}}(\mathcal{D}_{eval})$. The difficulty of the domain can be raised by increasing the number of objects. We sample datasets for 3 and 5 objects.

**Evaluation**: We compare against past works in compositional world modeling with publically available codebases at the time of writing. For the block pushing domain, we compare with homomorphic world models (GNN) (Zhao et al., 2022) and NPS (Goyal et al., 2021) with modifications to ensure that the actions are aligned with the slots (ALIGNEDNPS). The latter is equivalent to an ablation of COSMOS without the symbolic labelling module. We analyzed next-state predictions using mean squared error (MSE), current-state reconstructions using auto-encoder mean squared error (AE-MSE), and latent state separation using (MRR) following previous work (Kipf et al., 2019; Zhao et al., 2022; Goyal et al., 2021). Recognizing limitations in existing MRR calculation, we introduce the Equivariant MRR (Eq.MRR) metric, which accounts for different slot orders when calculating the MRR score. More details can be found in the appendix Section A.4.

**Results**: Results are presented in Table 2. First, we find that COSMOS achieves the best next state prediction performance (MSE) on all benchmarks. Moving from entity composition to relational composition datasets shows a drop in performance, underscoring the complexity of the task. Surprisingly, there is less than expected performance degradation moving from the three object to five

object domain. We attribute this to the nature of the block pushing domain and the choice of loss function. MSE loss measures the pixel error between the predicted and next image. As the density of objects in the image increases, the reconstruction target becomes more informative, encouraging better self-correction, and hence more efficient training.

Second, we observe that, without neurosymbolic grounding, the slot autoencoder's ability to encode and decode images degrades as the world model training progresses. This suggests that neurosymbolic grounding guards against auto-encoder representation collapse.

Finally, we do not notice a consistent pattern in the Equivariant MRR scores between models. First, all models tend to exhibit a higher Eq.MRR score in the five object environments. However, in many cases, models with high Eq.MRR score also have underperforming autoencoders. For instance, in the five object entity composition case, the GNN exhibits a high Eq.MRR score yet simultaneously has the worst autoencoder reconstruction error. We notice this happens when the model suffers a partial representation collapse (overfitting to certain color, shapes combination seen during training). This maps many slot encodings to the same neighborhood in latent space; making it easier to retrieve similar encodings, boosting the MRR score. Given these observations, we conclude that MRR might not be an optimal indicator of a model's downstream utility. For a comprehensive assessment of downstream utility, we turn to the methodology outlined by Veerapaneni et al. (2020), applying the models to a downstream planning task.

**Downstream Utility**: We evaluate the downstream utility of all world models using a simple planning task in 2D shapeworld environments. Our method, inspired by Veerapaneni et al. (2020), involves using a greedy planner that acts based on the Hungarian distance between the predicted and goal states. Due to the compounding nature of actions over a trajectory, there is an observed divergence from the ground truth the deeper we get into the trajectory. We run these experiments on our test dataset and average the scores at each trajectory depth. We showcase results in Figure 4. Our model (COSMOS) shows the most consistency and least deviation from the goal state in all datasets, which suggests that neurosymbolic grounding helps improve the downstream efficacy of world models. More details can be found in appendix section A.5.

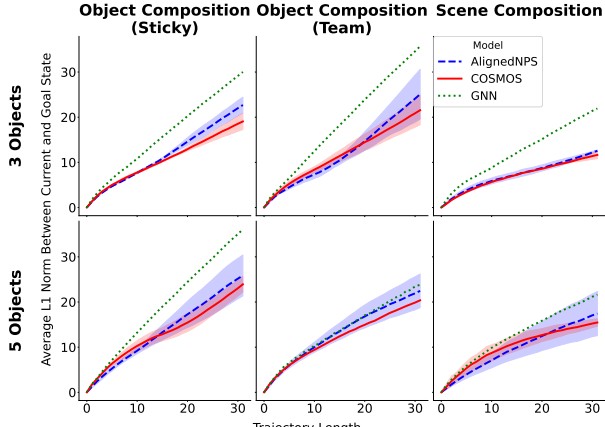

Figure 4: Downstream utility of different world models using a greedy planner. The graph follows the average L1 error between the chosen next state and the ground truth next state as a function of the number of steps the model takes. A lower L1 error indicates better performance. COSMOS (in red) achieves the best performance.

## 5 RELATED WORK

**Object-Oriented World Models**: Object-oriented world models (Kipf et al., 2019; Zhao et al., 2022; Van der Pol et al., 2020a; Goyal et al., 2021; Veerapaneni et al., 2020) are constructed to learn structured latent spaces that facilitate efficient dynamics prediction in environments that can be decomposed into unique objects. We highlight two primary themes in this domain.

*Interaction Modeling and Modularity in Representations*: Kipf et al. (2019) builds a contrastive world modeling technique using graph neural networks with an object-factored representation. However, GNNs may introduce compounding errors in sparse interaction domains. Addressing this, Goyal et al. (2021) introduces neural production systems (NPS) for modeling sparse interactions, akin to repeated dynamic instantiation GNN edge instantiation. COSMOS is influenced by the NPS architecture, with distinctions highlighted in Figure 2. In parallel, Chang et al. (2023) studies a hierarchical abstraction for world models, decomposing slots into a (dynamics relevant) state vector and a (dynamics invariant) type vector, with dynamics prediction focusing on the state vector. In

COSMOS, instead of maintaining a single "type vector", we maintain a set of learnable symbol vectors and select a relevant subset for each entity. This allows COSMOS to naturally discover global and local invariances, utilizing them to route latent encodings (akin to "state vectors") to appropriate transition modules.

*Compositional Generation and Equivariance in World Models*: Equivariant neural networks harness group symmetries for efficient learning (Ravindran, 2004; Cohen & Welling, 2016a;b; Walters et al., 2020; Park et al., 2022). In the context of CompGen, Van der Pol et al. (2020b) investigates constructing equivariant neural networks within the MDP state-action space. Zhao et al. (2022) establishes a connection between homomorphic MDPs and compositional generalization, expressing CompGen as a symmetry group of equivariance to object permutations and developing a world model equivariant to object permutation (using action attention). We adopt this idea, but integrate a modular transition model that also respects permutation equivariance in slot order.

**Vision-grounded Neurosymbolic Learning**: Prior work in neurosymbolic learning has demonstrated that symbolic grounding helps facilitate domain generalization and data efficiency (Andreas et al., 2016a;b; Sun et al., 2020; Zhan et al., 2021; Shah et al., 2020; Mao et al., 2019; Hsu et al., 2023; Yi et al., 2018). The interplay between neural and symbolic encodings in these works can be abstracted into three categories: (1) scaffolding perceptual inputs with precomputed symbolic information for downstream prediction (Mao et al., 2019; Ellis et al., 2018; Andreas et al., 2016b; Hsu et al., 2023; Valkov et al., 2018), (2) learning a symbolic abstraction from a perceptual input useful for downstream prediction Tang & Ellis (2023), and (3) jointly learning a neural and symbolic encodings leveraged for prediction Zhan et al. (2021). Our approach aligns most closely with the third category. While Zhan et al. (2021) combine neural and symbolic encoders in a VAE setting, highlighting the regularization advantages of symbols for unsupervised clustering, they rely on a program synthesizer to search for symbolic transformations in a DSL—introducing scalability and expressiveness issues. COSMOS also crafts a neurosymbolic encoding, but addresses scalability and expressiveness concerns of program synthesis by using a foundation model to generate the symbolic encodings.

**Foundation Models as Symbol Extractors**: Many works employ foundation models to decompose complex visual reasoning tasks (Wang et al., 2023a; Gupta & Kembhavi, 2023; Nayak et al., 2022). (Hsu et al., 2023; Wang et al., 2023b) decompose natural language instructions into executable programs using a Code LLM for robotic manipulation and 3D understanding. Notably, ViperGPT (Surís et al., 2023) uses Code LLMs to decomposes natural language queries to API calls in a library of pretrained models. Such approaches necessitate hand-engineering the API to be expressive enough to generalize to all attributes in a domain. COSMOS builds upon the idea of using compositionality of symbols to execute parameterized modules but sidesteps the symbolic decomposition bottleneck by parsimoniously using the symbolic encodings only for selecting representative encodings. I.e., COSMOS does not need its symbols to learn perfect reconstructions. The symbolic encoding is only used for selecting modules, while the neural encoding can learn fine-grained dynamics-relevant attributes that may not be known ahead of time. Furthermore, to the best of our knowledge, COSMOS is the first work to leverage vision-language foundation models for compositional world modeling.

## 6 CONCLUSION

We have presented COSMOS, a new neurosymbolic approach for compositionally generalizable world modeling. Our two key findings are that annotating entities with symbolic attributes can help with CompGen and that it is possible to get these symbols "for free" from foundation models. We have considered two definitions of CompGen — one new to this paper — and show that: (i) CompGen world modeling still has a long way to go regarding performance, and (ii) neurosymbolic grounding helps enhance CompGen.

Our work here aimed to give an initial demonstration of how foundation models can help compositional world modeling. However, foundation models are advancing at a breathtaking pace. Future work could extend our framework with richer symbolic representations obtained from more powerful vision-language or vision-code models. Also, our neurosymbolic attention mechanism could be naturally expanded into a neurosymbolic transformer. Finally, the area of compositional world modeling needs more benchmarks and datasets. Future work should take on the design of such artifacts with the goal of developing generalizable and robust world models.

# 7 REPRODUCIBILITY AND ETHICAL CONSIDERATIONS

## 7.1 REPRODUCIBILITY CONSIDERATIONS

We encourage reproducibility of our work through the following steps:

*Providing Code and Setup Environments*: Upon acceptance of our work, we will release the complete source code, pretrained models, and associated setup environments under the MIT license. This is to ensure researchers can faithfully replicate our results without any hindrance.

*Improving Baseline Reproducibility*: Our methodology extensively utilizes these public repositories (Zhao et al., 2022; Kipf et al., 2019; Goyal et al., 2021) for generating data and computing evaluation metrics. During development of our method, we made several improvements to these repositories: (1) We enhanced the performance and usability of the models, (2) We updated the algorithms to accommodate the latest versions of the machine learning libraries, and (3) we constructed reproducible anaconda environments for each package to work on Ubuntu 22.04. We intend to contribute back by sending pull requests to the repositories with our improvements, including other bug fixes, runtime enhancements, and updates.

*Outlining Methodology Details*: The details of our model are meticulously described in Section 3. A comprehensive overview is available in Algorithm 1, and a deeper dive into the transition function is in Appendix Section A.3. We mention training details in appendix section A.4.

## 7.2 ETHICAL CONSIDERATIONS

*Potential for Misuse*: As with other ML techniques, world models can be harnessed by malicious actors to inflict societal harm. Our models are trained on synthetic block pushing datasets, which mitigates their potential for misuse.

*Privacy Concerns*: In the long term, as world models operate on unsupervised data, they enable learning behavioral profiles without the active knowledge or explicit consent of the subjects. This raises privacy issues, especially when considering real-world, non-synthetic datasets. We did not collect or retain any human-centric data during the course of this project.

*Bias and Fairness*: World models, generally, enable learning unbiased representations of data. However, we leverage foundation models which could be trained on biased data and such biases can reflect in our world models.

# 8 ACKNOWLEDGEMENTS

This research was supported by the NSF National AI Institute for Foundations of Machine Learning (IFML), ARO award #W911NF-21-1-0009, and DARPA award #HR00112320018.

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

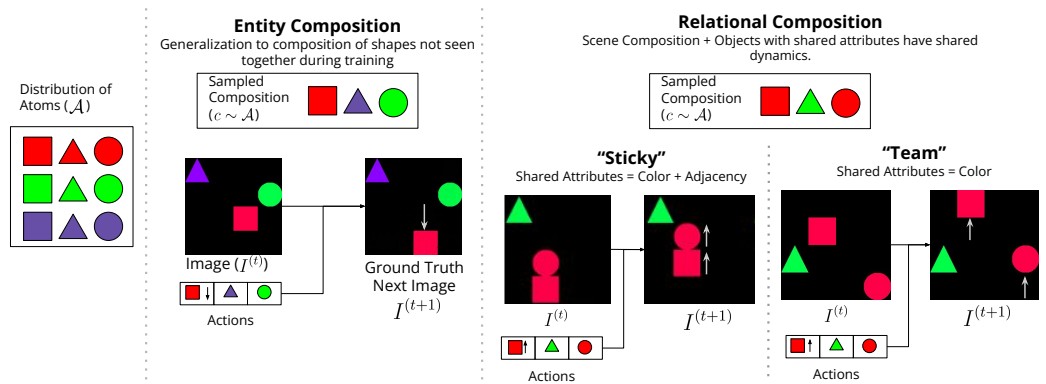

Figure 5: Overview of types of compositions studied. Entity composition (left) necessitates learning a world model that is equivariant to object replacement. Relational compositions (right) necessitates learning the properties of entity composition as well as additional constraints where objects with shared attributes also have shared dynamics. We study two instantiations of shared attributes sets: "Sticky" and "Team". Details on these instantiations are given in Appendix A.2.

# A  APPENDIX

The Appendix is divided into eight sections. Section A.1 explains how we leverage SAM to generate a set-structured representation, Section A.2 surveys the types of compositions we study in detail, Section A.3 introduces a faster algorithm used in implementation for module selection, Section A.4 outlines experiment details and, notably, presents justification for the Equivariant MRR metric employed to study encoding separation, Section A.5 presents details of how the downstream planning experiments were conducted, Section A.6 goes over our reasoning for selecting relevant benchmarks, Section A.7 details an ablation study with a "fully-symbolic" model, and Section A.8 showcases qualitative results on a randomly sampled subset of five-object state-action pairs.

## A.1  GENERATING SET-STRUCTURED REPRESENTATIONS WITH SEGMENT ANYTHING

We prompt SAM (Kirillov et al., 2023) with the image ($I$) and a $8 \times 8$ grid of points. This yields $64 \times 3$ potential masks (as there are three channels). To ensure a set structured representation, we must ensure that (1) each mask captures a specific property of the image, (2) collectively, all masks describe the entire image. We ensure (1) by removing duplicate masks and (2) by evaluating all combinations of remaining slots and selecting the $k$ tuple (where $k$ is the number of slots) that, when summed, most closely matches the image. The resulting masks $\{M'_1, \ldots M'_k\}$ are point-wise multiplied with the image to yield $\{I_1, \ldots I_k\}$. Each masked image is passed through a finetuned Resnet to yield $\{S_1, \ldots S_k\}$.

## A.2  TYPES OF COMPOSITIONS

**Entity Composition**: Entity composition (Figure 5) necessitates learning a world model that is equivariant to object replacement. The dynamics of the environment depends on which objects are present in the scene.

**Relational Composition**: Relational composition (Figure 5) necessitates learning all the properties present in entity composition. Additionally, in relational composition, the composition is determined by constraints placed on observable attributes of individual objects. For instance, in Sticky block pushing (Fig 5), the scene is constrained so that two objects start out with the same color adjacent to each other; and an action on one object moves all objects of the same color with it. This gives the appearance of two objects being stuck to each other. At test time, the objects stuck together change. Sticky block pushing demonstrates compositionality constraints based on two attributes: position and color. In the team block pushing (Figure 5), we relax the adjacency constraint in the sticky block pushing domain. An action on any object also moves other objects of the same color. This

allows us to study whether the adjacency constraint places a larger burden on dynamics learning than the color constraint.

## A.3 TRANSITION ALGORITHM

NPS (Goyal et al., 2021) necessitates selecting a primary slot $(p)$ to be modified, a contextual slot $(c)$, and a rule $(r)$ to modify the primary slot in the presence of the contextual slot. The naive algorithm to compute this tuple has a runtime of $O(k^2 l)$ where $k$ is the number of slots and $l$ is the number of rules. However, in implementation, the selection of $(r, p, c)$ can be reduced to a runtime of $O(kl + k)$ by *partial application* of the query-key attention. This is achieved by selecting the primary slot $p$ and $\text{MLP}_i$, partially transforming $S_p$ using a partial transition module $\text{MLP}_{(i,\texttt{left})}$, selecting the contextual slot $c$, and performing a final transformation of $\text{MLP}_{(i,\texttt{left})}(S_p)$ with $S_c$ using $\text{MLP}_{(i,\texttt{right})}$. Algorithm 2 presents this faster algorithm.

---

**Algorithm 2** Faster Transition Algorithm. This algorithm has a faster runtime than the one presented in the manuscript. The main difference is that the transition step is bifurcated into two parts, reducing the runtime of the selection from $O(k^2 l)$ to $O(kl + k)$ where $k$ is the number of slots and $l$ is the number of rules.

1: **function** TRANSITION(Key=$\{\overline{\Lambda}_1, \dots \overline{\Lambda}_k\}$, Query=$\{\vec{R}_1, \dots \vec{R}_l\}$, Value=$\{S_1, \dots S_k\}$)
2:  $\quad \mathbf{A}^\star \leftarrow \textbf{GumbelSoftmax}(\textbf{KQAttention}(\text{key}=\{\overline{\Lambda}_1, \dots \overline{\Lambda}_k\}, \text{query}=\{\vec{R}_1, \dots \vec{R}_l\}))$
3:  $\quad p, r \leftarrow \textbf{argmax}(\mathbf{A}^\star, \text{axis} = \texttt{`all'})$
4:  $\quad S^\star \leftarrow \text{MLP}_{r,\texttt{left}}(\text{concat}(S_p, \vec{R}_r))$
5:  $\quad \mathbf{A}_2^\star \leftarrow \textbf{GumbelSoftmax}(\textbf{KQAttention}(\text{key}=\{\overline{\Lambda}_1, \dots \overline{\Lambda}_k\}, \text{query}=\{S^\star, S^\star, S^\star\}))$
6:  $\quad c, \_ \leftarrow \textbf{argmax}(\mathbf{A}_2^\star, \text{axis} = \texttt{`all'})$
7:  $\quad S_2^\star \leftarrow \text{MLP}_{r,\texttt{right}}(\text{concat}(S_c, S^\star))$
8:  $\quad$ **return** $S_2^\star$

---

## A.4 EVALUATION PROCEDURE

**Dataset Generation**: To generate each dataset, we first create a scene configuration file that prescribes the permissible shapes and colors for objects within a given dataset. The scene configuration ensures that $\mathcal{F}_{\texttt{C}}(\mathcal{D}_{train}) \cap \mathcal{F}_{\texttt{C}}(\mathcal{D}_{eval}) = \emptyset$. Next, we sample $\mathcal{D}_{train}$ and $\mathcal{D}_{eval}$ from the permissible scenes. Each dataset is a trajectory of state-action pairs, where the state is the image of the shape $3 \times 224 \times 224$ and the action is a vector, factorized by object ID (Object x North-East-South-West). Overall, we generate 3000 trajectories of length 32 each where the actions are sampled from a random uniform distribution. Our domain is equivalent to the observed weighted shapes setup studied in Ke et al. (2021) with a compositionality constraint where the weight of the objects depend only on the shapes.

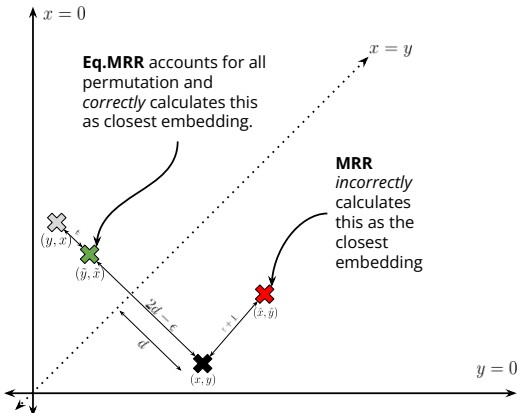

Figure 6: Intuition for shortcomings of MRR when number of slots $k = 2$ and $d_{slot} = 1$. The MRR metric incorrectly finds a point $(\hat{x}, \hat{y})$ that is $\epsilon + 1$ units away from $(x, y)$ while Equivariant MRR considers all possible permutations and finds a point $(\tilde{y}, \tilde{x})$ that is $\epsilon$ units away from $(y, x)$ and, in turn, closer to $(x, y)$ than $(\hat{x}, \hat{y})$.

**Baselines**: We compare against past works in compositional world modeling with publically available codebases at the time of writing. For the block pushing domain, we compare against homomorphic world models (GNN) (Zhao et al., 2022) and an ablation of our model without symbols (ALIGNEDNPS). GNN uses a slot autoencoder, an action attention module and a graph neural network for modeling transitions. It requires a two-step training process: first

| Dataset | Model | 3 objects | | | 5 objects | | |
|---|---|---|---|---|---|---|---|
| | | MSE ↓ | AE-MSE ↓ | Eq.MRR ↑ | MSE ↓ | AE-MSE ↓ | Eq.MRR ↑ |
| RC (Sticky) | COSMOS | **4.23E-03 +/- 1.49E-04** | **4.90E-04 +/- 1.03E-04** | **1.20E-01 +/- 2.05E-02** | **4.15E-03 +/- 3.21E-03** | **1.68E-03 +/- 1.48E-03** | **3.67E-01 +/- 7.73E-02** |
| | ALIGNEDNPS | 1.14E-02 +/- 9.89E-04 | 7.72E-03 +/- 1.21E-03 | 8.01E-02 +/- 6.79E-02 | 6.07E-03 +/- 8.30E-04 | 2.47E-03 +/- 3.60E-04 | 3.62E-01 +/- 1.81E-02 |
| | GNN | 7.94E-03 +/- 5.47E-03 | 5.11E-03 +/- 4.94E-03 | 6.03E-04 +/- 1.02E-04 | 6.21E-03 +/- 1.26E-03 | 2.73E-03 +/- 1.27E-03 | 5.30E-04 +/- 5.15E-05 |
| RC (Team) | COSMOS | **4.60E-03 +/- 2.32E-03** | **4.33E-04 +/- 1.58E-04** | 1.04E-01 +/- 3.19E-02 | **5.53E-03 +/- 1.95E-03** | 1.86E-03 +/- 1.61E-03 | 2.86E-01 +/- 4.32E-02 |
| | ALIGNEDNPS | 1.24E-02 +/- 4.11E-04 | 8.36E-03 +/- 6.78E-04 | **1.75E-01 +/- 2.68E-02** | 9.64E-03 +/- 1.95E-04 | 3.12E-03 +/- 6.07E-04 | **2.93E-01 +/- 2.02E-02** |
| | GNN | 8.92E-03 +/- 6.05E-03 | 3.82E-03 +/- 3.64E-03 | 7.16E-04 +/- 1.09E-04 | 7.01E-03 +/- 9.81E-04 | **1.62E-03 +/- 1.04E-03** | 5.46E-04 +/- 1.37E-04 |
| EC | COSMOS | **7.66E-04 +/- 4.08E-04** | **6.34E-05 +/- 2.01E-05** | 2.99E-01 +/- 2.85E-02 | **4.08E-04 +/- 4.68E-06** | **2.92E-06 +/- 6.34E-07** | 3.03E-01 +/- 3.88E-02 |
| | ALIGNEDNPS | 3.51E-03 +/- 6.30E-04 | 2.69E-03 +/- 6.89E-05 | 2.97E-01 +/- 7.99E-02 | 2.45E-03 +/- 3.47E-04 | 1.22E-03 +/- 9.06E-04 | 3.19E-01 +/- 1.01E-01 |
| | GNN | 9.89E-03 +/- 5.77E-03 | 1.03E-02 +/- 5.44E-03 | **5.50E-01 +/- 5.18E-01** | 1.20E-02 +/- 1.13E-02 | 1.28E-02 +/- 1.08E-02 | **5.25E-01 +/- 2.67E-01** |

Table 2: Evaluation results on the 2D block pushing domain for entity composition (EC) and relational composition (RC) averaged across three seeds. This table includes standard deviation numbers as well. Our model (COSMOS) achieves best next-state reconstructions for all datasets.

warm starting the slot-autoencoder and then training the action attention model and GNN with an equivariant contrastive loss (Hungarian matching loss). ALIGNEDNPS uses a slot autoencoder for modeling perceptions and a NPS module (Goyal et al., 2021) for modeling transitions. The pipeline is trained end to end with contrastive loss. We use action attention with NPS as well. For both models, we weren't able to reproduce the results using the provided codebases due to issues in training robust perception models for large images ($3 \times 224 \times 224$). To ensure a fair comparison, and since both these methods are agnostic to the perception model, we opt to reimplement the core ideas for both these models and use the same fine-tuned perception model for all models.

**Evaluation Procedure**: We evaluate all models on a single 48 GB NVIDIA A40 GPU with a (maximum possible) batch size of $64$ for 50 epochs for three random seeds. Contrastive learning necessitates a large batch size to ensure a diverse negative sampling set. As a result, the small batch size made contrastive learning challenging in our domain. To ensure a fair comparison, we report results for all models trained using reconstruction loss. We first train the slot autoencoder (ENTITYEXTRACTOR and SPATIALDECODER) until the model shows no training improvement for 5 epochs. This is sufficient to learn slot autoencoders with near-perfect state reconstructions. All transition models are initialized with the same slot-autoencoder and are optimized to minimize a mixture of the autoencoder reconstruction loss and the next-state reconstruction loss. For compositional world modeling, we are interested in two aspects of model performance: next-state predictions and separation between latent states. We evaluate next-state predictions on all models using the mean squared error (MSE) between the predicted next image and the ground truth next image in the experience buffer. We also measure the performance of the autoencoder on reconstructing the current state by calculating the slot-autoencoder mean squared error (AE-MSE). Generally, training world models improves the perception model's ability to reconstruct states as well. We also evaluate the separability of the learned latent encodings. This is done by measuring the L2 distance between the predicted next slot encodings and the ground-truth next slot encodings obtained from the encoder and using information theoretic measures such as mean reciprocal rank (MRR) to measure similarity. Notably, the MRR computation in previous work does not to account for the non-canonical order of slots, causing higher L2 distance and, consequently, higher MRR scores when the target and predicted slots have different orderings. The core issue here is that MRR computation, as used in previous works, fixed the order of the slots before calculating L2 distance. This ignored $k! - 1$ possible orderings where a closer target encoding could be found. To rectify this, we propose a new metric, Equivariant MRR (Eq.MRR), which uses the minimum L2 distance among all permutations of slot encodings to calculate mean reciprocal rank. This metric ensures that the latent slot encodings are not penalized for having different slot orders. Figure 6 presents an illustration of the shortcomings of MRR on a simple example. This limitation is characteristic of algorithms which do not align the slots to a canonical slot ordering. In practice, we observe that the Equivariant MRR is always lower than or equal to the MRR.

## A.5 DOWNSTREAM EVALUATION SETUP

Following Veerapaneni et al. (2020), we use a greedy planner that chooses the action that minimizes the Hungarian distance between the current and the goal state. These actions are applied $t - 1$ times over a trajectory of length $t$, with the output from the world model at the $(d - 1)$-th step becoming the state for step $d$ in the trajectory. Due to this compounding nature, we see an increased divergence from the ground truth as we get deeper into the trajectory. At each step $d$ in the trajectory,

| Dataset | Only Symbols (MSE ↓) | Only Neural / AlignedNPS (MSE ↓) | COSMOS (MSE ↓) |
|---|---|---|---|
| 3 Object RC - Sticky | 1.36E-02 | 1.14E-02 | **4.23E-03** |
| 3 Object RC - Team | 1.39E-02 | 1.24E-02 | **4.60E-03** |
| 3 Object EC | 1.21E-02 | 3.51E-03 | **7.66E-04** |

Table 3: Evaluation results on the 2D block pushing domain for ablations of COSMOS.

the accuracy of the world model is evaluated as the L1 error of the difference between the current ground truth and predicted states in the form of their $xy$-coordinates. These $xy$-coordinates are initialized for each object to the origin and updated with every action taken by the corresponding rule. For example, after one step, if the ground truth moves an object to the east, but the planner chooses to move the same object to the west, then the distance between the two states would be 2. We run these experiments for the 500 trajectories of length $t = 32$ in our test dataset and average the scores at each trajectory depth. We showcase results in Figure 4. Our model (COSMOS) shows the most consistency and least deviation from the goal state in all datasets, which suggests that neurosymbolic grounding helps improve the downstream efficacy of world models.

## A.6 DATASET COMPARISON

The focus of our paper is to demonstrate the first neuro-symbolic framework leveraging foundation models for compositional object-oriented world modelling, and we evaluate on the same benchmarks as existing work (Kipf et al., 2019; Goyal et al., 2021; Zhao et al., 2022). We chose our evaluation domain based on four properties: (1) object-oriented state and action space, (2) history-invariant dynamics, (3) action conditioned (plannable) dynamics, and (4) ease of generating new configurations (to evaluate entity and relational composition). We curate the following list of domains from related work to explain what properties are missing for each dataset.

| Dataset and Relevant Works | Object Oriented State and Action Space | History Independent | Action Conditioned (Plannable) | Configurable |
|---|---|---|---|---|
| 2D Block Pushing (Kipf et al., 2019; Goyal et al., 2021) | | | | |
| (Zhao et al., 2022; Ke et al., 2021) | ✓ | ✓ | ✓ | ✓ |
| Physion (Wu et al., 2023) | ✓ | | | ✓ |
| Phyre (Wu et al., 2023) | ✓ | | | ✓ |
| Clevrer (Wu et al., 2023) | ✓ | | | |
| 3DObj (Wu et al., 2023) | ✓ | | | ✓ |
| MineRL (Hafner et al., 2023) | | | ✓ | |
| Atari (Pong/Space invaders/Freeway) (Goyal et al., 2021) | ✓ | | | |
| Minigrid/BabyAI Mao et al. (2022) | ✓ | | ✓ | ✓ |

## A.7 SYMBOLIC ABLATION OF COSMOS

ALIGNEDNPS serves as a "fully neural" ablation to demonstrate the effectiveness of our model. In this section, we detail another "fully symbolic" ablation of our model to demonstrate the need for a neurosymbolic approach. Specifically, we maintain the algorithm presented in 1 but modify the transition model to use the symbolic embedding to predict the next state. Specifically, line 9 changes to:

$$S_p \leftarrow S_p + \text{MLPBANK}[r](\textbf{concat}(\Lambda_p, \ \Lambda_c, \ \vec{R}_r))$$

The results, detailed in Table 3, indicate that the "symbols-only" model significantly underperforms compared to COSMOS. We believe this is because the symbolic embedding is constructed by concatenating symbolic attributes, and the rule module is not aware of this structure. This causes the MLP to overfit to the attribute compositions seen at train time. COSMOS sidesteps this issue by using the symbolic embedding in the key-query attention module to select the relevant rule module, while allowing the real vector to learn local features useful for modeling action-conditioned transitions.

## A.8 QUALITATIVE RESULTS

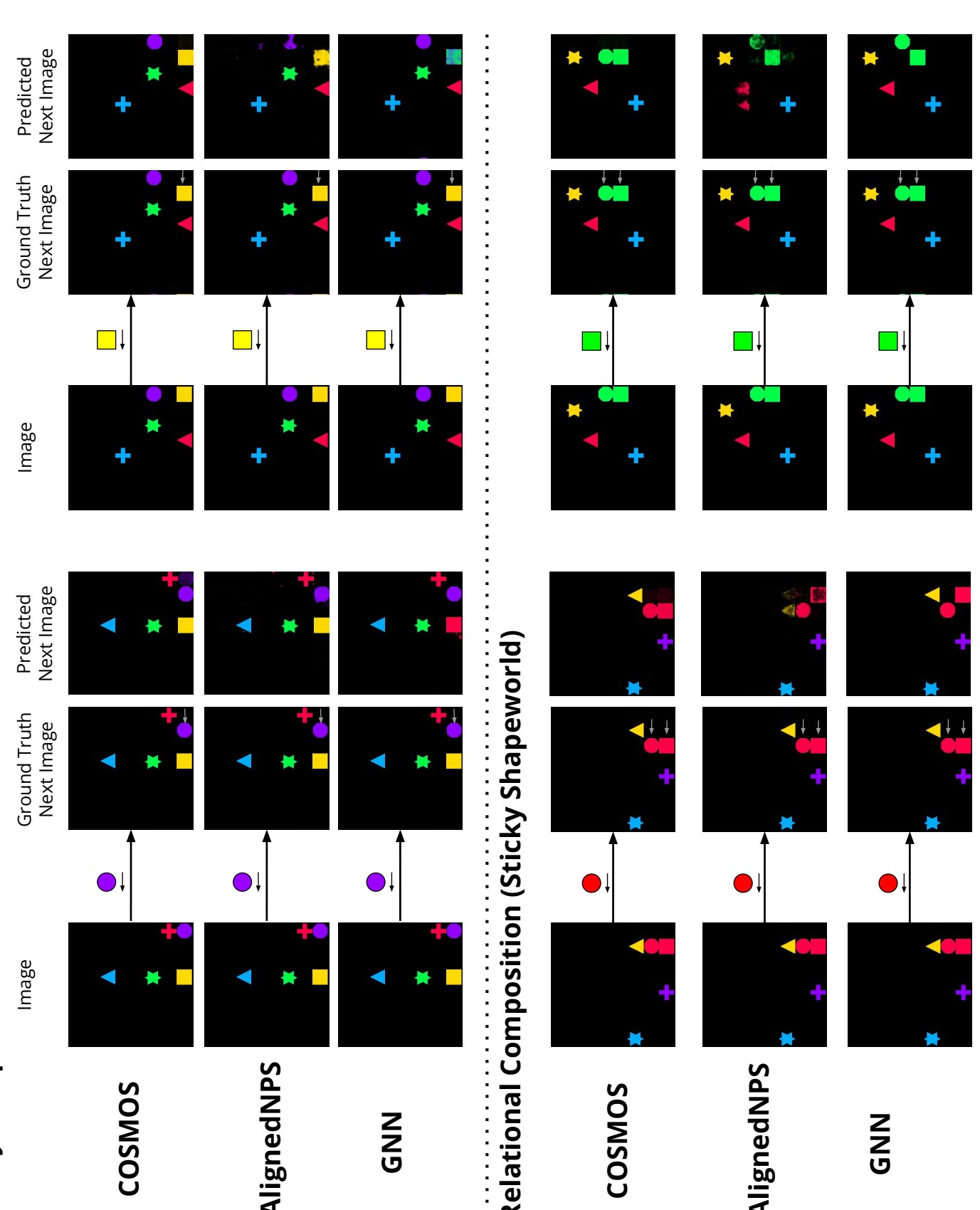

Figure 7: Qualitative outputs on randomly chosen state–action pairs for all baselines. We show two samples for each experiment and dataset type with 5 objects. Color is shown for illustrative purposes only; in implementation, the action conditioning does not carry any information about color.

