# OpenReview forum: "Neurosymbolic Grounding for Compositional World Models"
_ICLR.cc/2024/Conference — ICLR 2024 poster_

### Official Review · Reviewer_QCEm · 2023-10-26

**Soundness:** 3 good
**Presentation:** 3 good
**Contribution:** 2 fair
**Rating:** 6
**Confidence:** 4

**Summary:**

The authors proposed a framework for object-centric world modeling in compositional generalization settings. They propose a neuro-symbolic scene encoding, consisting of real vectors and vectors of symbols describing attributes, as well as a neuro-symbolic attention mechanism, which binds entities to rules of interaction. They use foundation models to extract each entity’s symbolic attributes, and show their method’s performance on a block-pushing domain.

**Strengths:**

I am appreciative of the neural-symbolic attention mechanism, computed between ordered symbolic and neural rule encodings to determine the most applicable rule-slot pair. I also appreciate that this is end-to-end differentiable as a permutation equivariant action attention module. Using a frozen foundation model to capture the symbolic attributes in the scene encodings is also a simple but elegant way of decomposing entities into attributes.

**Weaknesses:**

W1. I am not convinced that the proposed neuro-symbolic scene encoding is the optimal formulation. I would have appreciated ablations in this paper, including ones that explored whether or not having both real vectors and a vector of symbolic attributes is actually helpful. What should the real vector capture that the symbolic attribute is not capturing, especially given that the rules of the evaluation domain seem to only rely on these attributes?

W2. An additional note is that I would have appreciated more clarity of what information is given to the model, and whether other methods have access to the same information. From my understanding, a strong assumption is that the symbolic labeling module requires a predefined list of attributes that is important for the downstream task. I think in many downstream tasks this may not be reasonable to know a priori. Potentially, you can run experiments showing that with a superset of attributes, directly predicted by some foundation model, that you can still learn the correct correspondence to rules given this noise.

W3. Similarly, are there assumptions made on how many rules there are in the evaluation domain? I understand that the rule is a learnable encoding, but is the amount of rules learned as well? One can imagine that the method discovers rules that are correct, but not optimal (e.g., decompose rules into many smaller rules that overfit to the train set).

W4. I would have appreciated evaluation on a different domain, such as maybe Physion, and learn more complex and less obvious rules such as rigid and soft-body collisions, stable multi-object configurations, etc. In the block-pushing domain, it seems like the rules are tied to clear attributes such as shape and color, without having to further learn whether non-uniform combinations of these would lead to certain downstream effects. Also related to the known attributes assumption in W2.

**Questions:**

Q1. Can you clarify how SAM is used with ResNet to produce a set-structured hidden representation for each object?

Q2. Is there a way to interpret the learned rules and qualitatively see how well it aligns with the ground truth rules?

---

> ### Author Response · Authors · 2023-11-16
> **Response to reviewer QCEm**
>
> Thank you for reading our paper in detail. We address concerns inline:
>
> > *I am not convinced that the proposed neuro-symbolic scene encoding is the optimal formulation. I would have appreciated ablations in this paper, including ones that explored whether or not having both real vectors and a vector of symbolic attributes is actually helpful. What should the real vector capture that the symbolic attribute is not capturing, especially given that the rules of the evaluation domain seem to only rely on these attributes?*
>
> We appreciate your insightful comment regarding the neuro-symbolic scene encoding. In our paper (Section 4, Evaluation), we included the AlignedNPS baseline as a “fully neural” ablation to demonstrate the effectiveness of our model. Based on your suggestion, we have now conducted additional experiments with a “fully symbolic” ablation. The results, detailed below, indicate that the “symbols-only” model significantly underperforms compared to COSMOS. We believe this is because the symbolic embedding is constructed by concatenating symbolic attributes, and the rule module is not aware of this structure. This causes the MLP to overfit to the attribute compositions seen at train time. COSMOS sidesteps this issue by using the symbolic embedding in the key-query attention module to select the relevant rule module, while allowing the real vector to learn local features useful for modeling action-conditioned transitions. We will update our writing in Section 4, so this is more clear.
>
> | **Dataset**          | **Only Symbols (MSE)** | **Only Neural / AlignedNPS (MSE)** | **COSMOS (MSE)** |
> | -------------------- | ---------------------- | ---------------------------------- | ---------------- |
> | 3 Object OC - Sticky | 1.36E-02               | 1.14E-02                           | **4.23E-03**     |
> | 3 Object OC - Team   | 1.39E-02               | 1.24E-02                           | **4.60E-03**     |
> | 3 Object SC          | 1.21E-02               | 3.51E-03                           | **7.66E-04**     |
>
> > *W2. An additional note is that I would have appreciated more clarity of what information is given to the model, and whether other methods have access to the same information. From my understanding, a strong assumption is that the symbolic labeling module requires a predefined list of attributes that is important for the downstream task. I think in many downstream tasks this may not be reasonable to know a priori. Potentially, you can run experiments showing that with a superset of attributes, directly predicted by some foundation model, that you can still learn the correct correspondence to rules given this noise.*
>
> We use a closed vocabulary with 6 colors and 5 shapes that we used to generate the dataset. These colors and shapes were used for all models. We appreciate the suggestion from the reviewer, and ran the suggested experiment with a superset of attributes. Specifically, we increased the number of colors to 10 colors and the number of shapes to 9 shapes. Due to compute and time limitations during the rebuttal, we ran this for one experimental configuration where we retrained an open vocabulary ablation of our model for a fixed number of epochs (50). The preliminary results indicate that the performance remains comparable even with the expanded set of attributes. This finding suggests that our model can adapt to a broader range of attributes without significant loss in performance. We will include these discussions in the final version of the paper in Section 4.
>
> |                                      | COSMOS(Validation MSE) | COSMOS-OpenVocab(Validation MSE) |
> | ------------------------------------ | ---------------------- | -------------------------------- |
> | OC-Sticky 3 objects(After 50 epochs) | 6.17E-3                | 6.03E-3                          |
>
>
>
> > *W3. Similarly, are there assumptions made on how many rules there are in the evaluation domain? I understand that the rule is a learnable encoding, but is the amount of rules learned as well? One can imagine that the method discovers rules that are correct, but not optimal (e.g., decompose rules into many smaller rules that overfit to the train set).*
>
> The number of rules is fixed ahead of time and was selected based on what empirical performance - we use 5 rules and 3 application steps. Each rule embedding is paired with a rule module. These rule modules are implemented as MLP’s and have the capacity to model many action/slot interactions.
>
> *(continued in next thread)*

---

> > ### Author Response · Authors · 2023-11-16
> > **Response to reviewer QCEm (continued; part 1)**
> >
> > >  *W4. I would have appreciated evaluation on a different domain, such as maybe Physion, and learn more complex and less obvious rules such as rigid and soft-body collisions, stable multi-object configurations, etc. In the block-pushing domain, it seems like the rules are tied to clear attributes such as shape and color, without having to further learn whether non-uniform combinations of these would lead to certain downstream effects. Also related to the known attributes assumption in W2.*
> >
> >
> >
> > The focus of our paper is to demonstrate the first neuro-symbolic framework leveraging foundation models for compositional object-oriented world modelling, and we evaluate on the same benchmarks as existing work `[1, 2, 3]` (at ICLR 2019, NeurIPS 2021, ICML 2022 respectively). We chose our evaluation domain based on four properties: (1) object-oriented state and action space, (2) history-invariant dynamics, (3) action conditioned (plannable) dynamics, and (4) ease of generating new configurations (to evaluate scene and object composition). We curate the following list of domains from related work to explain what properties are missing for each dataset:
> >
> > | **Dataset**                         | Relevant Works  | **Object Oriented state and action space** | **History Independent** | **Action Conditioned (Plannable)** | **Configurable** |
> > | ----------------------------------- | --------------- | ------------------------------------------ | ----------------------- | ---------------------------------- | ---------------- |
> > | 2D Block Pushing                    | `[1, 2, 3,  4]` | ✅                                          | ✅                       | ✅                                  | ✅                |
> > | Physion                             | `[5]`           | ✅                                          | ❌                       | ❌                                  | ✅                |
> > | Phyre                               | `[5]`           | ✅                                          | ❌                       | ❌                                  | ✅                |
> > | Clevrer                             | `[5]`           | ✅                                          | ❌                       | ❌                                  | ❌                |
> > | 3DObj                               | `[5]`           | ✅                                          | ❌                       | ❌                                  | ✅                |
> > | MineRL                              | `[6]`           | ❌                                          | ❌                       | ✅                                  | ❌                |
> > | Atari (Pong/Space invaders/Freeway) | `[2]`           | ✅                                          | ❌                       | ❌                                  | ❌                |
> > | Minigrid/BabyAI                     | `[7]`           | ✅                                          | ❌                       | ✅                                  | ✅                |
> >
> > Specifically, we do not evaluate on  Physion/Phyre/etc. as these datasets are not action conditioned and we cannot do planning in them. Models for such domains condition next state generation using a series of burn-in frames `[5]`. While we cannot evaluate COSMOS directly on such datasets, we can reuse parts of our architecture (the neurosymbolic attention module and the symbolic labelling module) in a transformer based architecture such as the one demonstrated in  `[5]` (ICLR, 2023). Preliminary results show better reconstruction loss performance than slotformer on the CLEVRER dataset. Note we do not evaluate on CLEVRER initially, since this is a video prediction dataset like Physion/Phyre, and we cannot do planning. Furthermore, we cannot evaluate generalization w.r.t object compositionality in this dataset as the video generation scripts are not available.
> >
> > | Algorithm            | Validation MSE after 5k steps ($\downarrow$) |
> > | -------------------- | -------------------------------------------- |
> > | COSMOS + Transformer | **8.024E-3**                                 |
> > | `[5]` (ICLR,2023)    | 8.975E-3                                     |
> >
> > *(continued in next thread)*

---

> ### Author Response · Authors · 2023-11-16
> **Response to reviewer QCEm (continued; part 2)**
>
> >  *Q1. Can you clarify how SAM is used with ResNet to produce a set-structured hidden representation for each object?*
>
> We prompt SAM with the image ($I$) and a 8x8 grid of points. This yields 64*3 potential masks (as there are three channels). To ensure a set structured representation, we must ensure that (1) each mask captures a specific property of the image, (2) collectively, all masks describe the entire image. We ensure (1) by removing duplicate masks and (2) by evaluating all combinations of remaining slots and selecting the $k$ tuple (where $k$ is the number of slots) that, when summed, most closely matches the image. The resulting masks $\{M_1, M_2\dots M_k\}$ are point-wise multiplied with the image to yield $\{I_1, I_2, \dots I_k\}$. Each image is passed through a Resnet pretrained on Imagenet with the final layer finetuned to yield $\{S_1, S_2, \dots S_k\}$. We will add these implementation details to the Appendix.
>
> >  *Q2. Is there a way to interpret the learned rules and qualitatively see how well it aligns with the ground truth rules?*
>
> We don’t expect the rules to be interpretable or correspond to ground truth rules because the rule modules are implemented as MLPs and usually encode many state/action transitions. The modular architecture can be utilized to facilitate a more interpretable design – for instance, by using linear layers instead of MLPs for rule modules. We leave this direction on interpretability for future work.
>
> **References**
>
> - `[1]`: http://arxiv.org/abs/1911.12247 (ICLR 2019)
> - `[2]`: https://arxiv.org/abs/2103.01937 (NeurIPS 2021)
> - `[3]`: https://arxiv.org/abs/2204.13661 (ICML 2022)
> - `[4]`: https://arxiv.org/abs/2107.00848 (ICLR 2021)
> - `[5]`: https://arxiv.org/abs/2210.05861 (ICLR 2023).
> - `[6]`: https://arxiv.org/abs/2301.04104 (Preprint)
> - `[7]`: https://arxiv.org/abs/2303.05501 (NeurIPS 2022)

---

> > ### Comment · Reviewer_QCEm · 2023-11-20
> > **Thanks for the response**
> >
> > Thanks for the thoughtful response! I think it is promising that the open vocabulary ablation seems to yield comparable performance, though I would like to see the final results. On the other hand, I believe that fixing the number of rules ahead of time is a limitation, and limits the application of this method to real world tasks. Hence, I am keeping my score as is.

---

### Official Review · Reviewer_CQ89 · 2023-10-31

**Soundness:** 3 good
**Presentation:** 3 good
**Contribution:** 2 fair
**Rating:** 6
**Confidence:** 4

**Summary:**

This paper presents a neurosymbolic approach to learning a transition model in the pixel space. The proposed model, namely Cosmos, takes in the current state as an image, the action (encoded as one-hot vectors), and predicts the next state. The model is trained on a dataset of state-action-state transition tuples and evaluated on unseen state-action combinations. The model focuses on two types of compositional generalization tests: scene composition and object composition.

**Strengths:**

This paper tackles an important problem that is interesting to the ICLR community, specifically learning world models from pixels. The overall presentation of the paper is good. The organization of the method section clearly illustrates the number of modules in the system and how they are connected with each other, which is obviously helpful for readers. The description of experimental setups is clear, and the authors have done a sufficient number of ablation studies and comparisons with baselines.

**Weaknesses:**

There are two main weaknesses of the paper.

First, the problem setting of object composition seems very contrived to me, for two reasons.
- In the physical world, it is unclear what's a concrete example where such kind of metarules would apply. In particular, the authors are training the model on seeing two red blocks moving together and two green blocks moving together, and hope that the model would generalize to predict two blue blocks would also move together. Such kind of "attribute-relationship" based generalization doesn't seem natural to me. Arguably, this kind of generalization can be dangerous: two blocks can be stacked together; two cylinders can be stacked together, but not two spheres, in the physical world.
- There is some serious machine learning identifiability issue with this setting. If the model does not have inductive biases in training, there is no way that it can generalize.
--- Based on these two concerns, the arguments around object compositional generalization is weak.

Second, the model is only trained and evaluated on a fairly toy environment, and the downstream application to planning is only shown in a very simple setting. It is unclear how this approach can be generalized to more complicated scenarios.

Slightly minor is that the paper missed some important related work along the direction of learning neuro-symbolic transition models. For example,
- PDSketch: Integrated Planning Domain Programming and Learning https://arxiv.org/abs/2303.05501
- Learning Object-Oriented Dynamics for Planning from Text https://openreview.net/forum?id=B6EIcyp-Rb7

They are not exactly the same setting but a lot of the high-level ideas are definitely the same, including learning lifted transition rules, using factorized embeddings (colors, shapes, etc.) to represent objects.

While I overall like the presentation of the paper---it's well-organized and overall good. I found the description of some details very unclear. In particular:
- Page 3, the object composition part. I have to check the appendix and read through paragraphs/figures several times in order to understand what this object composition means. I think the name is not very descriptive. The authors should consider change it to a better name that describes such kind of "metarules" (e.g., two objects have relations if they share the same color) and present concrete examples in the main text.
- Parge 4-5: the authors should keep the "..." in the sets. Otherwise it's very confusing to look at "{c1, c2, cp}"
- The writing of the method section could be further improved by having a running example (and referring back to this example in 3.1, 3.2, and 3.3).

Finally, the paper does not have a limitation discussion section.

Minor notes on the writing: I think using CG as an acronym for "compositional generalization" is a bit uncommon. The term is easily confused with other concepts like "computer graphics."

**Questions:**

I don't have particular questions. Please address the missing related works; and consider reframing and better illustrating the object compositional generalization.

---

> ### Author Response · Authors · 2023-11-14
> **Response to reviewer CQ89**
>
> Thank you for reading our paper in detail. We address concerns inline:
>
> > *The problem setting of object composition seems very contrived to me.*
>
> Our definition of “object compositionality” refers to compositions of objects in any setting where entity dynamics are dependent on attributes of other entities. This problem setting commonly occurs in real-world scenarios (ex: object stacking, team sports, autonomous driving, modeling almost any type of agent interactions, etc.). On the other hand, scene compositionality is where entity dynamics are independent of the dynamics of other entities. We evaluate on the same benchmark as existing work `[1, 2, 3]` (at ICLR 2019, NeurIPS 2021, ICML 2022 respectively) and choose our evaluation domain based on four properties: (1) object-oriented state and action space, (2) history-invariant dynamics, (3) action conditioned (plannable) dynamics, and (4) ease of generating new configurations (to evaluate scene and object composition). See the detailed response below.
>
> >  *In the physical world, it is unclear what's a concrete example where such kind of metarules would apply…Such kind of "attribute-relationship" based generalization doesn't seem natural to me.*
>
> There are many instances where composition of objects are governed by observable attributes. For instance, “team composition” naturally occurs in sports (soccer players wearing the same colored jersey move in similar ways), autonomous driving (yellow taxi cabs share macro-behaviors), object manipulation tasks (color-coded gas cylinders or color-coded wires must be handled in the same way, etc.) “sticky composition” occurs naturally in pick-and-place tasks.
>
> > *In particular, the authors are training the model on* **[(1)]** *seeing two red blocks moving together and two green blocks moving together, and hope that the model would generalize to predict two blue blocks would also move together. Arguably, this kind of generalization can be dangerous:* **[(2)]** *two blocks can be stacked together; two cylinders can be stacked together, but not two spheres, in the physical world*
>
> We don’t study this scenario since it requires the model to **generalize to new attributes not seen during training** (in (1) this is color, in (2) this is 3D shape). In our object compositionality scenarios, we are training the model to **generalize across entity combinations, with each attribute/entity already seen during training time**. For instance, at training time in (1), the model might observe all shape-color pairs and compositions of (circle, triangle) / (triangle, square) sharing the same color move in unison. At test time, we expect the model to work for a scenario where (circle, square) share the same color.
>
> In scenario (2) brought up by the reviewer, in our definition of “sticky” object compositionality, we enforce that each object occurs at least once on top and once on the bottom during training. Therefore, the model would always have seen a sphere/cone at the bottom; allowing it to learn the rule that no object can be stacked on top of a “sphere/cone”. Our definition necessitates that the distribution of atoms doesn’t shift across data splits (ie: all shape-color combinations are seen at training time). We will clarify Section 2 so this is more clear, and see a more detailed discussion below (relational compositonality section).
>
> >  *There is some serious machine learning identifiability issue with this setting. If the model does not have inductive biases in training, there is no way that it can generalize. --- Based on these two concerns, the arguments around object compositional generalization is weak.*
>
> We make sure that our dataset is constructed so that for every input image and action, there is a deterministic one-to-one function that produces the next image. This holds for the scene composition and the object composition datasets. We ensure that the distribution of atoms is fixed between training and test sets, while the distribution of compositions has no overlap between sets. This ensures that the model has seen all objects with a particular shape-color at train time, but the particular composition of objects it sees at train and test time vary. The compositions placed in train and test distributions are shuffled evenly as well. (Appendix A.3)
>
> *(continued in next thread)*

---

> ### Author Response · Authors · 2023-11-14
> **Response to reviewer CQ89 (continued; part 1)**
>
> > *Second, the model is only trained and evaluated on a  fairly toy environment, and the downstream application to planning is only shown in a very simple setting. It is unclear how this approach can be generalized to more complicated scenarios.*
>
> The focus of our paper is to demonstrate the first neuro-symbolic framework leveraging foundation models for compositional object-oriented world modelling, and we evaluate on the same benchmarks as existing work `[1, 2, 3]` (at ICLR 2019, NeurIPS 2021, ICML 2022 respectively). We chose our evaluation domain based on four properties: (1) object-oriented state and action space, (2) history-invariant dynamics, (3) action conditioned (plannable) dynamics, and (4) ease of generating new configurations (to evaluate scene and object composition). We curate the following list of domains from related work to explain what properties are missing for each dataset:
>
> | **Dataset**                         | Relevant Works  | **Object Oriented state and action space** | **History Independent** | **Action Conditioned (Plannable)** | **Configurable** |
> | ----------------------------------- | --------------- | ------------------------------------------ | ----------------------- | ---------------------------------- | ---------------- |
> | 2D Block Pushing                    | `[1, 2, 3,  4]` | ✅                                          | ✅                       | ✅                                  | ✅                |
> | Physion                             | `[5]`           | ✅                                          | ❌                       | ❌                                  | ✅                |
> | Phyre                               | `[5]`           | ✅                                          | ❌                       | ❌                                  | ✅                |
> | Clevrer                             | `[5]`           | ✅                                          | ❌                       | ❌                                  | ❌                |
> | 3DObj                               | `[5]`           | ✅                                          | ❌                       | ❌                                  | ✅                |
> | MineRL                              | `[6]`           | ❌                                          | ❌                       | ✅                                  | ❌                |
> | Atari (Pong/Space invaders/Freeway) | `[2]`           | ✅                                          | ❌                       | ❌                                  | ❌                |
> | Minigrid/BabyAI                     | `[7]`           | ✅                                          | ❌                       | ✅                                  | ✅                |
>
> Following the reviewer’s suggestions, to demonstrate the application of our framework in more complex domains, we also present preliminary results on CLEVRER utilizing a transformer based framework similar to `[5]` (ICLR, 2023). Note that CLEVRER is a video prediction dataset like Physion/Phyre, so we cannot do planning on this dataset. Furthermore, we cannot evaluate generalization w.r.t object compositionality in this dataset as the video generation scripts are not available.
>
> | Algorithm            | Validation MSE after 5k steps ($\downarrow$) |
> | -------------------- | -------------------------------------------- |
> | COSMOS + Transformer | **8.024E-3**                                 |
> | `[5]` (ICLR,2023)    | 8.975E-3                                     |
>
> With respect to the downstream application to planning, we do not claim to develop SOTA planning methods for world models. Our experiment aims to evaluate the downstream efficacy of our world models using off the shelf planning algorithms leveraged by related work `[8]`.
>
> >  *Slightly minor is that the paper missed some important related work along the direction of learning neuro-symbolic transition models.*
>
> Thank you for the references. We shall include these works in the final manuscript. Some comments on each work:
>
> - PDSketch: PDSketch’s representation consists of symbols lifted from images using pretrained models, which are expressed using tensors (neural relaxations of variables; similar to NS-CL cited in Section 5.2). The problem with such a representation is that the symbols become a bottleneck for representation learning. We sidestep this bottleneck by parsimoniously using the symbolic encodings only for selecting representative encodings. More information is in Section 5.3.
>
> - Learning OOD for Planning from Text: OOTD also learns an object-oriented latent representation and learns a neural network to predict the next state conditioned on an action. There seem to be two big architectural differences: 1) COSMOS uses key-query attention to locally updates selected objects, while OOTD uses a GNN to globally update all objects simultaneously, and 2) OOTD doesn’t use any symbolic grounding.
>
> *(continued in next thread)*

---

> ### Author Response · Authors · 2023-11-14
> **Response to reviewer CQ89 (continued; part 2)**
>
> > *I think the name [object composition] is not very descriptive. The authors should consider a better name that describes such kind of "metarules" (e.g., two objects have relations if they share the same color) and present concrete examples in the main text.*
>
> Thank you for pointing this out. We initially chose this name to describe compositionality over levels of scenes vs. objects, and we agree with the reviewer that another name would be more clear. We plan to revise the naming convention based on the independence w.r.t dynamics.
>
> - *Entity composition (previously named scene composition)*: The dynamics of individual entities are independent of the dynamics of other entities.
> - *Relational composition (previously named object composition)*: The entity dynamics are dependent/related to attributes of other entities.
>   - Team composition: shared attribute is color
>   - Sticky composition: shared attribute is color+adjacency
>
> We really appreciate your feedback, and we would be happy to further clarify this if needed.
>
> >  *The authors should keep the "..." in the sets. Otherwise it's very confusing to look at "{c1, c2, cp}."*
>
> We shall make these changes. Thank you for the suggestions!
>
> >  *The writing of the method section could be further improved by having a running example (and referring back to this example in 3.1, 3.2, and 3.3).*
>
> Great suggestion. We shall incorporate the example used in Figure 3 in section 3.1, 3.2, 3.3, etc. if space permits.
>
> >  *Finally, the paper does not have a limitation discussion section.*
>
> Due to space limitations, we decided to discuss limitations and future work together in Section 6, last paragraph. We plan to add an explicit limitations section to the appendix. The two main limitations of our method are:
>
> 1. Dependence on CLIP: The symbol grounding depends on the quality of the foundation model. However, foundation models are an exciting area with rapid developments - as a result, we expect the performance of our framework to improve as new methods are released.
> 2. More benchmarks and datasets: As the reviewer has pointed out, there is a lack of datasets with real world physics interactions. In an earlier response, we show preliminary results on CLEVRER (a block pushing dataset with collision physics) demonstrating that our framework can be used to handle more complex domains.
>
> >  *I think using CG as an acronym for "compositional generalization" is a bit uncommon.*
>
> We initially chose this for brevity, but agree with the reviewer that it could be confusing. We will refer to compositional generalization as “CompGen.” We are also happy to discuss other suggestions.
>
> **References**
>
> - `[1]`: http://arxiv.org/abs/1911.12247 (ICLR 2019)
> - `[2]`: https://arxiv.org/abs/2103.01937 (NeurIPS 2021)
> - `[3]`: https://arxiv.org/abs/2204.13661 (ICML 2022)
> - `[4]`: https://arxiv.org/abs/2107.00848 (ICLR 2021)
> - `[5]`: https://arxiv.org/abs/2210.05861 (ICLR 2023).
> - `[6]`: https://arxiv.org/abs/2301.04104 (Preprint)
> - `[7]`: https://arxiv.org/abs/2303.05501 (NeurIPS 2022)
> - `[8]`: https://arxiv.org/abs/1910.12827 (CoRL 2019)

---

> ### Comment · Reviewer_CQ89 · 2023-11-20
> **Response to Authors**
>
> I appreciate the responses from the authors. I have also read reviews from other reviewers. I decided to increase my score by 1.

---

### Official Review · Reviewer_5cti · 2023-11-12

**Soundness:** 2 fair
**Presentation:** 3 good
**Contribution:** 3 good
**Rating:** 6
**Confidence:** 2

**Summary:**

This paper proposed COSMOS framework for compositional generalization in world modeling, that involves a blend of neural and symbolic representations to understand scene entities and their interactions.

**Strengths:**

1.     This paper proposes an end-to-end differentiable framework with a novel neuro-symbolic attention
2.	This is a well written and well-structured paper.
3.	Most of the traditional neuro-symbolic methods map the representations to symbol manually while this paper does it without any manual effort

**Weaknesses:**

1.	The effectiveness of the framework might be constrained dealing with larger and more complex real-world scenarios. Since scalability of neuro-symbolic methods are required to handle more diverse environments.
2.	When the model will encounter with noisy or incomplete input, how will the model perform?
3.	Combining neural and symbolic inputs might be computationally heavy, but there is no significant discussion about the computational complexity.
4.	Although the model showcases strong performance in the 2D block pushing domain with MSE but that’s not the case for MRR and Eq.MRR always. More experimental results are required to establish this as the new state-of-the-art.

5. There are typos in introduction section.

**Questions:**

point 2 from weakness section

---

> ### Author Response · Authors · 2023-11-14
> **Response to reviewer 5cti**
>
> Thank you for reading our paper in detail. We address concerns inline:
>
> > *The effectiveness of the framework might be constrained dealing with larger and more complex real-world scenarios. Since scalability of neuro-symbolic methods are required to handle more diverse environments.*
>
> The focus of our paper is to demonstrate the first neuro-symbolic framework leveraging foundation models for compositional object-oriented world modelling, and we evaluate on the same benchmarks as existing work `[1, 2, 3]` (at ICLR 2019, NeurIPS 2021, ICML 2022 respectively). We chose our evaluation domain based on four properties: (1) object-oriented state and action space, (2) history-invariant dynamics, (3) action conditioned (plannable) dynamics, and (4) ease of generating new configurations (to evaluate scene and object composition). We curate the following list of domains from related work to explain what properties are missing for each dataset:
>
> | **Dataset**                         | Relevant Works  | **Object Oriented state and action space** | **History Independent** | **Action Conditioned (Plannable)** | **Configurable** |
> | ----------------------------------- | --------------- | ------------------------------------------ | ----------------------- | ---------------------------------- | ---------------- |
> | 2D Block Pushing                    | `[1, 2, 3,  4]` | ✅                                          | ✅                       | ✅                                  | ✅                |
> | Physion                             | `[5]`           | ✅                                          | ❌                       | ❌                                  | ✅                |
> | Phyre                               | `[5]`           | ✅                                          | ❌                       | ❌                                  | ✅                |
> | Clevrer                             | `[5]`           | ✅                                          | ❌                       | ❌                                  | ❌                |
> | 3DObj                               | `[5]`           | ✅                                          | ❌                       | ❌                                  | ✅                |
> | MineRL                              | `[6]`           | ❌                                          | ❌                       | ✅                                  | ❌                |
> | Atari (Pong/Space invaders/Freeway) | `[2]`           | ✅                                          | ❌                       | ❌                                  | ❌                |
> | Minigrid/BabyAI                     | `[7]`           | ✅                                          | ❌                       | ✅                                  | ✅                |
>
> To demonstrate the application of our framework in more complex domains, we also present preliminary results on the CLEVRER dataset utilizing a transformer based framework similar to `[5]` (ICLR, 2023). Note that CLEVRER is a video prediction dataset, so we cannot do planning on this dataset. Furthermore, we cannot evaluate generalization w.r.t object compositionality in this dataset as the video generation scripts are not available.
>
> | Algorithm            | Validation MSE after 5k steps ($\downarrow$) |
> | -------------------- | -------------------------------------------- |
> | COSMOS + Transformer | **8.024E-3**                                 |
> | `[5]` (ICLR,2023)    | 8.975E-3                                     |
>
> Furthermore, traditional neuro-symbolic representation learning approaches searched for programs to generate the required symbols in a combinatorial search space `[8]`. The reviewer correctly points out that such search methods do not scale as the complexity of the environment increases. **We do not suffer from these scalability issues** as we use a foundation model to query symbols. The foundation model encodes the program in its parameter space, and inference has a constant time complexity. While we use CLIP for our experiments, in principle, we are agnostic to the choice of vision-language foundation model, allowing us to generalize to different domains.
>
> > *When the model will encounter with noisy or incomplete input, how will the model perform?*
>
> Symbols generally enable the model to be more robust to noisy and incomplete inputs. The CLEVRER domain referred in the previous response has noisy inputs in the form of partial occlusion. Preliminary results show that our model performs better than a comparable baseline.
>
> *(continued in next thread)*

---

> ### Author Response · Authors · 2023-11-14
> **Response to reviewer 5cti (continued)**
>
> >  *Combining neural and symbolic inputs might be computationally heavy, but there is no significant discussion about the computational complexity.*
>
> The transition module has the highest computational complexity and dictates the complexity of the overall algorithm. We mention the computational complexity of our transition module in Section 3.3. Specifically, it has a $\mathcal{O}(kl + k)$ time complexity, where $k$ is the number of slots and $l$ is the number of rules.
>
> Unlike previous work `[8]` (TMLR, 2022) which necessitates searching for programs in a combinatorial space, we query symbolic attributes using a foundation model. Since each symbolic attribute query is independent of other queries, we can batch all queries together. Consecutively, the symbolic labelling module has a constant time complexity and $\mathcal{O}(kp)$ space complexity, where $p$ is the number of classes. Typically, $k$ is larger than $p$.
>
> > *Although the model showcases strong performance in the 2D block pushing domain with MSE but that’s not the case for MRR and Eq.MRR always. More experimental results are required to establish this as the new state-of-the-art.*
>
> We offer an explanation for the performance inconsistencies of Eq.MRR in (Section 4, Results). Specifically, we notice that models with high MRR tend to have underperforming autoencoders. For instance, in the five object scene composition case, the GNN exhibits a high Eq.MRR score yet simultaneously has the worst autoencoder reconstruction error. We notice this happens when the model suffers a partial representation collapse (overfitting to certain color, shapes combination seen during training). This maps many slot encodings to the same neighborhood in latent space; making it easier to retrieve similar encodings, boosting the MRR score.
>
> The appendix section on the shortcomings of MRR and Eq.MRR further elucidates issues with using this metric for compositional world modeling (A.3, Evaluation Procedure). We are happy to discuss this point further if there are remaining questions.
>
> > *There are typos in introduction section.*
>
> Thank you for pointing this out. I believe this is “consistsd” which should be “consists.” We will update the introduction accordingly.
>
>
>
> **References**
>
> - `[1]`: http://arxiv.org/abs/1911.12247 (ICLR 2019)
> - `[2]`: https://arxiv.org/abs/2103.01937 (NeurIPS 2021)
> - `[3]`: https://arxiv.org/abs/2204.13661 (ICML 2022)
> - `[4]`: https://arxiv.org/abs/2107.00848 (ICLR 2021)
> - `[5]`: https://arxiv.org/abs/2210.05861 (ICLR 2023).
> - `[6]`: https://arxiv.org/abs/2301.04104 (Preprint)
> - `[7]`: https://arxiv.org/abs/2303.05501 (NeurIPS 2022)
> - `[8]`: https://arxiv.org/abs/2107.13132 (TMLR, 2022)

---

> > ### Comment · Reviewer_5cti · 2023-11-22
> > **response to author's rebuttal**
> >
> > I express gratitude to the authors and acknowledge their responses to my reviews. My concerns have been adequately addressed, and I am maintaining my initial decision.

---

### Meta-Review · Area_Chair_auAS · 2023-12-07

**Metareview:**

This paper presents an end-to-end differentiable neurosymbolic framework to learn a transition model in the pixel space. The proposed model, namely Cosmos, takes in the current state as an image, the action (encoded as one-hot vector), and predicts the next state.  There are two technical components: (i) neurosymbolic scene encodings, which represent each entity in a scene using a real vector and a vector of composable symbols describing attributes of the entity, and (ii) a neurosymbolic attention mechanism that binds these entities to learned rules of interaction. The reviewers generally find the problem interesting, the proposed framework technically solid (neural-symbolic attention, end-to-end differentiable, etc), and the work well presented.   In the meantime, they have some concerns on whether the method can be generalized to more complicated (large and noisy) scenarios.

**Justification For Why Not Higher Score:**

No reviewer is excited about this work.

**Justification For Why Not Lower Score:**

It is reasonable.

---

### Decision · Program_Chairs · 2024-01-16

Accept (poster)